# GramGAN: Deep 3D Texture Synthesis From 2D Exemplars

**Tiziano Portenier[1], Siavash Bigdeli[2], Orcun Goksel[1]**
1: Computer-assisted Applications in Medicine, ETH Zurich, Switzerland
2: Swiss Center for Electronics and Microtechnology, Switzerland
`{tiziano.portenier, orcun.goksel}@vision.ee.ethz.ch`
`siavash.bigdeli@csem.ch`

## Abstract

We present a novel texture synthesis framework, enabling the generation of infinite, high-quality 3D textures given a 2D exemplar image. Inspired by recent advances in natural texture synthesis, we train deep neural models to generate textures by non-linearly combining learned noise frequencies. To achieve a highly realistic output conditioned on an exemplar patch, we propose a novel loss function that combines ideas from both style transfer and generative adversarial networks. In particular, we train the synthesis network to match the Gram matrices of deep features from a discriminator network. In addition, we propose two architectural concepts and an extrapolation strategy that significantly improve generalization performance. In particular, we inject both model input and condition into hidden network layers by learning to scale and bias hidden activations. Quantitative and qualitative evaluations on a diverse set of exemplars motivate our design decisions and show that our system performs superior to previous state of the art. Finally, we conduct a user study that confirms the benefits of our framework.

## 1 Introduction

Texture synthesis is an important field of research with a multitude of applications in both computer vision and graphics. While vision applications mainly focus on texture analysis and understanding, in computer graphics textures are most important in the field of image manipulation and photo-realistic rendering. In this context, applications require synthesis techniques that can generate highly realistic textures. Due to the problem being highly complex, today's applications largely rely on techniques that generate textures based on real photos, which is a time-consuming process that requires expert knowledge. In this work, we present a novel texture synthesis framework that can generate high-quality 3D textures given an exemplar image as input (See Figure 1).

Techniques for texture synthesis can be classified into photogrammetric and procedural techniques. Photogrammetry relies on methods that capture the color appearance of physical objects to obtain (typically several) photos from the target texture. Next, texture mapping algorithms are employed to supply a digital 3D model, usually a polygon mesh, with texture coordinates that define a mapping from 3D surface points to 2D texture pixels. Such photogrammetric approaches yield high-quality textures but rely on time-consuming procedures that require expert knowledge. Moreover, the required projection from 3D to 2D introduces seam and distortion artifacts, for example when texturing spherical objects. Nevertheless, this is still a widely used technique, despite its shortcomings.

An entirely different class of techniques is referred to as *procedural texturing*, where textures are generated by an algorithm instead of relying on acquired data. Ideally the algorithm takes a 3D position as input and computes a texture color value as output, omitting the error-prone projection step entirely. Pioneering work in procedural texturing is the Perlin noise texture model [1], which

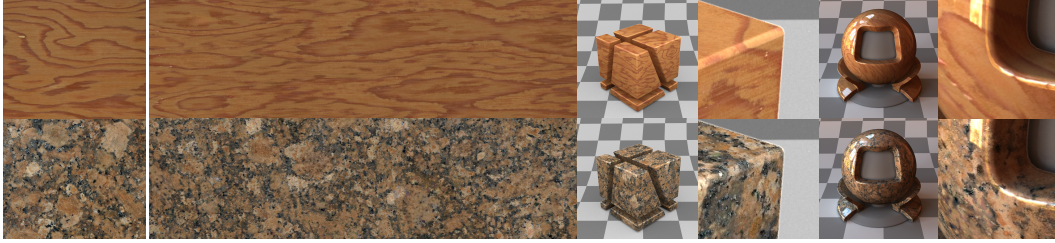

Figure 1: Results produced by our framework. Given an exemplar texture image (left), our model can faithfully resynthesize it infinitively (middle), suitable to texture 3D models (right).

dates back to 1985. The core idea is to linearly combine noise frequencies, or octaves, to generate different textures. An artist manually adjusts octave weights to obtain a specific look. While this early approach conveniently avoids both data acquisition and texture mapping, its applicability to natural textures is rather limited mainly due to the predefined, fixed noise frequencies. Moreover, a user has to manually tweak the parameters in order to obtain a desired look, which is a tedious process.

Recently, several techniques that leverage deep learning have been proposed to train generative models for procedural texture synthesis. Most of these leverage convolutional neural networks (CNNs) to generate texture pixel values as a function of spatially neighboring data. Recent examples in this direction either use generative adversarial networks (GANs) [2] or a style loss [3] to optimize their models. While these CNN-based approaches yield high-quality textures, in practice their application is limited to either 2D textures or very low resolution 3D textures due to the high memory footprint of 3D CNNs. Moreover, an unsolved problem with these techniques is the lack of diversity or rather *stochasticity*, i.e. such models struggle producing diverse results that represent the same texture. Despite attempts to mitigate this problem by explicitly minimizing a diversity term [4], true diversity has not been demonstrated by this approach.

Our work is inspired by to the model proposed by Henzler et al. [5], which features both memory efficiency and diversity. The former is achieved by formulating the texture synthesis problem as a point operation parametrized using a multilayer perceptron (MLP), while the latter is achieved by defining the model to be a function of noise frequencies, similarly to Perlin. However, this flexibility comes at the cost of lower image quality and a lack of similarity to the exemplar texture.

We present a point operation model that learns to generate textures as a function of noise frequencies, but we propose a novel loss function and architectural framework, which together improve image quality significantly. Moreover, our proposed network architecture inherently supports an extrapolation technique that improves generalization to unseen textures, that can be substantially different from the training set exemplars. In particular, we make the following contributions: (1) a deep MLP architecture that effectively combines learned noise frequencies; (2) a novel GAN loss inspired by style transfer to achieve both realistic image quality and high similarity to the input exemplar; and (3) a novel extrapolation strategy that significantly improves the results on unseen exemplars.

## 2 Related work

Texture synthesis techniques have a long history in computer vision and graphics. In 1985, Perlin [1] proposed a model to generate stochastic textures by linearly combining frequencies of noise in 2D or 3D. By using different combination weights, various visual appearances can be created. However, due to fixed, isotropic frequencies, the model expressiveness is limited and the space of generated textures is restricted to isotropic textural features, inhibiting anisotropic, e.g. elongated structures. Another major limitation of this model in practice is that the linear weights must be tediously tweaked to obtain an output that matches a desired texture. To overcome this limitation, a desired property of any texture synthesis method is the ability to generate textures that automatically match an exemplar texture at hand. Early works that tackle this problem are patch-based techniques [6–11], where texture exemplars are either padded, inpainted, or reproduced from scratch by considering local neighborhoods in the exemplar image. While these methods can faithfully reproduce a target texture, they are limited to 2D and are computationally expensive. With recent advances in deep learning, a multitude of learned texture synthesis models have been proposed in the vision community. Most

such work either optimize a style loss [3], a GAN loss [2], or a hybrid of the two. As our work is closely related to these approaches, we give an overview of the respective methods in the following.

**Texture synthesis using style loss**   In their groundbreaking work, Gatys et al. [3] demonstrated that statistical correlations of activations extracted from a pre-trained VGG-19 network [12] encode textural features. In particular, it has been shown that matching the Gram matrices of these deep activations successfully guides the synthesis network to generate samples similar to the target texture. Building upon these insights, several methods have been proposed to improve computation time [13, 14] and diversity [4] of texture generation. All such models are parametrized using CNNs and therefore they do not scale well to 3D textures at acceptable texture resolutions. Moreover, these approaches all work on a single exemplar texture, which implies the need for training an individual model for each target texture. Finally, obtaining true diversity in generated samples is still an unsolved problem with these methods. A big leap forward has been achieved recently by Henzler et al. [5]. They propose a point operation model parametrized as an MLP instead of using CNNs, which enables efficient synthesis of high-resolution 3D textures. Moreover, true diversity is demonstrated thanks to combining noise frequencies directly, similarly to Perlin noise textures. Finally, their model is capable of synthesizing an entire space of textures, rather than a single exemplar, when trained on a set of exemplar textures. However, the resulting texture quality is not yet on par with state of the art CNN models and the synthesized textures are often relatively different from the respective target exemplars. In our work, we also model point operations that combine noise frequencies to efficiently support diverse 3D texture synthesis. However, we show that our novel GAN loss and network architecture significantly improve image quality and similarity to target textures. Moreover, we demonstrate superior generalization performance enabled by our proposed network architecture.

**Texture synthesis using GANs**   GANs are another widely used class of models for texture synthesis, by training a CNN generator $G$ to fool a CNN discriminator $D$ in an adversarial minimax game [15–18]. Most such approaches learn purely generative models, often trained on individual exemplars. An effective technique to incorporate a user-provided exemplar is conditional GANs [19], as shown by Alanov et al. [20]. Here, $G$ synthesizes a texture given an exemplar patch as input. To enforce $G$ to respect its input, $D$ discriminates the joint distribution of pairs of texture patches. Here, a sample of the generated distribution consists of a condition patch and a synthesized texture, and a sample of the ground truth distribution consists of two random patches coming from the same texture. A more widely used approach to incorporate a conditional input is to extend the generator loss by an additional style loss term [3] using an off-the-shelf feature extractor (VGG) [21–23]. While GAN-based models achieve astonishing texture quality, even surpassing style-loss based techniques [15], the proposed models all rely on CNNs, hindering their applicability to 3D textures due to computational limitations. While we also leverage GANs in our work to obtain high-quality textures, we propose a model based on point operations, suitable for 3D texture synthesis. Moreover, we incorporate ideas from style-loss based techniques to train a conditional GAN without the need of an additional off-the-shelf network, resulting in a unified loss function that is much more efficient than hybrid losses.

## 3   GramGAN

In this section we introduce our framework for 3D texture synthesis from 2D exemplars. At the core of our method is a texture sampler that is modeled as an MLP to efficiently learn to combine learned noise frequencies using a specialized layer architecture (Figure 2). We train this model by optimizing a novel loss function that incorporates concepts from both GANs and style loss approaches to achieve high-quality outputs. In this work we consider two different target applications: (1) learning to synthesize highly realistic textures from a single exemplar (Section 3.1), and (2) learning a texture latent space given a set of many exemplars (Section 3.2). Later in Section 3.3 we propose an efficient strategy for improving the generalization performance.

### 3.1   Efficient, infinite, and highly realistic 3D textures from a single 2D exemplar

Given an exemplar image of a target texture, in order to learn a model that can generate new, high-quality samples of the target texture in 3D, we train a transformation model and a texture sampler. Our transformation model learns to produce $n$ noise frequencies, or octaves $T_i, i \in [0, n-1]$, given the 2D exemplar. These frequencies are linear 3D transformations $T_i \colon \mathbb{R}^3 \to \mathbb{R}^3$ defined as $T_i c = c_i$

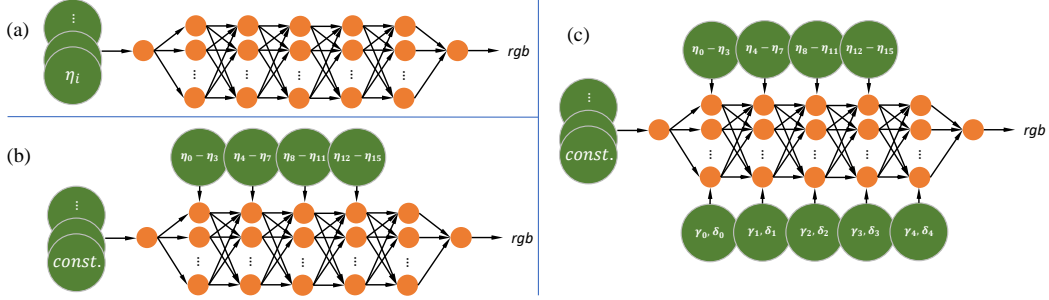

Figure 2: Sampler network models. a) The baseline model, where the noise is inserted at the input layer. b) Our model that combines blocks of noise frequencies at hidden layers (Equation 2). c) Our conditional model that combines the conditional input at each hidden layer (Equation 5).

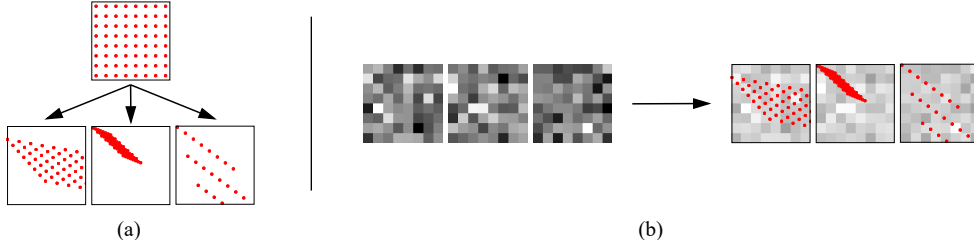

Figure 3: 2D example of coordinate transformations and noise sampling using 3 octaves. a) Input coordinates $c$ (top) are transformed using learned frequencies $T_i$ (bottom). b) Noise instances $\bar{\eta}$ (left) are trilinearly sampled using transformed coordinates (right).

that map a spatial 3D coordinate $c$ to a set of new locations $\{c_i\}$, one location for each frequency. See Figure 3 (a) for an illustration of a 2D example using 3 octaves.

The sampler $S$ maps the spatial coordinate $c$ to its RGB texture value. We first compute a noise value $\eta_i(c)$ for each octave by trilinearly sampling $n$ 3D Gaussian noise tensor instances $\bar{\eta}_i \sim \mathcal{N}(0,1)$ using the transformed locations $c_i$, i.e., $\eta_i(c) = \bar{\eta}_i(c_i)$ (Figure 3 (b)). This results in an input noise vector $\eta(c) = (\eta_0(c), \eta_1(c), ..., \eta_{n-1}(c))$ that serves as input to $S$ for computing the final RGB output at location $c$:

$$S\colon \mathbb{R}^n \to \mathbb{R}^3, \quad S\big(\eta(c)\big) = \text{RGB}(c). \tag{1}$$

During training, we simultaneously optimize both the parameters of $S$ and the $n \times 3 \times 3$ matrix entries in the transformation model.

We parametrize $S$ using an MLP with 5 hidden layers, as shown in Figure 2. We find that directly providing the transformed noise $\eta$ at the input layer yields subpar outputs. Therefore, we propose to inject $\eta$ into the hidden layers of $S$ and learn a constant input layer instead. Consider a dense layer defined as $y = \phi(Wx + b)$, where $W$ is the weight matrix, $b$ is the bias vector, $x$ is the layer input, $\phi$ is the activation function, and $y$ stores the layer activations. We use dense layers of the form:

$$y = \phi(Wx + \psi(\eta) + b) \quad \text{with} \quad \psi(\eta) = A\eta, \tag{2}$$

where $A$ is a learned $m \times n$ matrix for a layer with $m$ hidden units. This concept resembles some similarities to StyleGAN [24], although the input noise has a different role here and network layers of our sampler do not relate to scale in any way. In practice we find it beneficial to force individual layers in $S$ to specialize on a subset of noise frequencies. This reduces the number of parameters to optimize as well as minimizing the risk of $S$ ignoring certain frequencies in computation of its output. In particular, we partition the frequencies in equal bins of size $n/4$ and inject each bin into one hidden layer only. The last hidden layer thus does not receive any noise, which we find to reduce the risk of ignoring other frequencies further, and the dimension of the matrices $A$ reduces to $m \times n/4$.

**Loss function** We optimize the parameters of our model using a GAN loss. However, while our sampler generates 3D textures, our critic $D$, parametrized as a standard CNN, discriminates 2D

images. In particular, in each training iteration our model synthesizes 2D images by randomly slicing the 3D space. To optimize $D$, we use the WGAN-GP loss proposed by Gulrajani et al. [25], i.e.,

$$\mathcal{L}_D = \mathbb{E}[D(f)] - \mathbb{E}[D(r)] + \lambda \mathbb{E}[(||\nabla_u D(u)||_2 - 1)^2], \qquad (3)$$

where $f$ is a randomly oriented 2D slice produced by our model, $r$ is a random patch from the exemplar texture image, and $u$ is a data point uniformly sampled along the straight line connecting $r$ and $f$. This is a crucial difference to [5], where only axis-aligned slices are sampled during training. To optimize our texture generator $G$, we introduce an additional style term $\mathcal{L}_{\text{style}}$ to the WGAN loss:

$$\mathcal{L}_G = -\alpha \mathbb{E}[D(f)] + \beta \mathcal{L}_{\text{style}} \quad \text{with} \quad \mathcal{L}_{\text{style}} = \frac{1}{L} \sum_{l=0}^{L-1} \frac{1}{4N_l^2 M_l^2} \sum_{i,j} |M_{ij}^l(F(r)) - M_{ij}^l(F(f))|, \quad (4)$$

where $L$ is the number of hidden layers in a feature extractor network $F$, $M^l(F(x))$ is the Gram matrix of the activations in layer $l$ when presenting $x$ as input to $F$, $N_l$ is the number of filters in layer $l$, and $M_l$ is the number of pixels in the vectorized activations of layer $l$. Note that while previous work on style loss employ pretrained, off-the-shelf feature extractors (typically $F = $ VGG-19 [3, 13, 14, 4, 5, 21–23]), our important difference is that we set $F = D$, i.e., we train $F$ simultaneously with $G$ in an adversarial manner! This concept has numerous advantages compared to the traditional hybrid approaches based on combinations of GAN and style losses. First, the activations in $D$ are anyhow computed during GAN training, hence our approach significantly minimizes any computational overhead in loss function evaluation. Second and more importantly, such features explicitly trained for texture discrimination, as we show later, help generate textures of significantly higher quality compared to off-the-shelf features trained on image recognition as in works leveraging VGG. Note that we minimize $L_1$ distance between Gram matrices in $\mathcal{L}_{\text{style}}$ as opposed to the $L_2$ distance commonly used in related work, since we find this to improve the stability of adversarial training. Note that even though the concept of replacing VGG features with discriminator features has been proposed in the literature before in the context of perceptual losses, leveraging the discriminator in a style loss context has not been studied before.

## 3.2   Learning a latent texture space

While training a distinct texture synthesis model for each target texture is computationally feasible, in practice it is desirable to have a single conditional model that can generate an entire class, or space, of textures. Besides the reduced training computation time, such a model promises generalization capabilities such as interpolation and extrapolation, greatly simplifying the texture generation process from the user perspective. Our goal is to obtain a texture synthesis framework that allows a user to select a 2D image patch of a target texture, and the model can generate new samples of this target texture in 3D without retraining. For this purpose we extend our model proposed in Section 3.1 by an encoder $E(x) = z$ that maps an input exemplar patch $x$ to a compact latent representation $z$. Similar to Henzler et al. [5], we parametrize $E$ using a CNN. The idea is now to condition both $T_i$ and $S$ on the latent code $z$. Note that, thanks to our novel loss function, we train an unconditional GAN (i.e., an unconditional discriminator but a conditional generator). This is a crucial feature in texture synthesis; conditioning a discriminator on a different patch coming from the same exemplar texture is problematic since natural textures often feature significant low-frequency variations. Our proposed loss function guarantees that the generator cannot ignore the condition without directly involving the discriminator.

In order to incorporate the latent code $z$ in the synthesis process, we train an additional 3-layer MLP $Q(z) = \{T_i\}$ to map the latent representation $z$ to a set of noise frequencies $\{T_i\}$. Moreover, we extend $S$ to non-linearly combine noise frequencies given $z$, i.e., $S(\eta(c)|z) = \text{RGB}(c)|z$. Instead of presenting the condition $z$ at the input layer of $S$, we find it beneficial to directly manipulate the hidden activations in $S$ as a function of $z$ (See Figure 2). Our approach is inspired by attention modules [26] and adaptive instance normalization techniques [27] for CNNs. First, we train a small 3-layer MLP $f$ to learn a texture style $w = f(z)$. Next, each hidden layer $l$ learns an affine mapping $g_l(w) = (\gamma, \delta)$ that computes a scale and bias for each unit in the respecitve layer. The definition from Equation 2 therefore extends to

$$y = \gamma(\phi(Wx + \psi(\eta) + b)) + \delta \quad \text{with} \quad \gamma, \delta \in \mathbb{R}^m. \qquad (5)$$

Note that in order to synthesize a texture sample given an input exemplar, a single forward pass through $E$, $Q$, $f$, and $g_l$ is required. Only $S$ is evaluated for each RGB value to be synthesized.

### 3.3 Extrapolation to unseen data

While our proposed conditional model generalizes well to unseen samples that are sufficiently similar to the training data and interpolation in latent space produces plausible textures, we find that extrapolation, the generalization performance to entirely new textures, leaves a lot to be desired. This is not surprising since generalization in the sense of extrapolation is a major unsolved problem for deep learning models. Nevertheless, supported implicitly by the architectural design of our texture sampler $S$, we propose an extrapolation strategy that significantly improves texture quality on exemplars not belonging to the training data manifold. Our method is a mixture between zero-shot learning and domain-adaption.

Given a new exemplar texture patch $x$, our idea is to fine-tune only a fraction of the parameters of our model in order to help improve the quality of generated samples. For this purpose we use $E$ and $Q$ to predict an initial guess for the latent code $z$ and the transformations $\{T_i\}$. Next, we estimate a texture style $w$ and from it a scale and bias vector $\gamma, \delta$ for each hidden layer in $S$. In addition, we also predict the noise weight matrix $A$ for the first four hidden layers in $S$. Finally, we perform stochastic gradient descent by evaluating random slices of the generated texture in each iteration. During this optimization we solely alter the parameters $\theta = (T_i, \gamma_l, \delta_l, A_l)$, with $l$ being the layer index in $S$, while keeping all other parameters fixed. While also keeping the parameters in $D$ fixed, we minimize $\mathcal{L}_{\text{style}}$ for a few hundred iterations.

## 4 Experiments and results

In this section we evaluate the effect of the components of our framework by ablating them and we compare our system with a state of the art 3D texture synthesis method [5]. Note that all synthesized 2D images shown in this work are in fact slices in 3D. For more results, comparisons, and analysis, we refer the reader to the supplemental material.

**Implementation details** We train all models using Adam optimizer [28] and we use (equalized [29]) learning rates of $2 \times 10^{-3}$ for $D$ and $5 \times 10^{-4}$ for $G$ ($E$, $Q$, and $S$). In each training iteration both $G$ and $D$ are updated once and sequentially. In $\mathcal{L}_D$ we set $\lambda = 10$. An important factor is the choice of the hyperparameters $\alpha$ and $\beta$ in $\mathcal{L}_G$. Although various settings produce plausible outputs, we achieved best results when setting $\alpha = 0.1$ and $\beta = 1$. Note that training a conditional model is not feasible unless $\beta > 0$. Setting $\alpha = 0$ leads to perfectly stable adversarial training in both the single exemplar and the conditional setting, which we find a rather surprising insight. In the single exemplar setting we set $n = 16$ and our conditional models use $n = 32$ noise frequencies. We train on texture patches of size $128^2$ and use noise instances of resolution $64^{3n}$. Note that the choice of texture patch resolution decides on the *receptive field* of both the discriminator $D$ and the texture encoder $E$, i.e., what textural structure size the model can capture. Using larger or smaller texture patches only requires the CNN architectures of $E$ and $D$ to be modified accordingly, the sampler architecture is not affected by this choice. $E$ maps texture patches to a latent representation of dimension 32. Our MLPs $Q$, $S$, and $f$ consist of 1, 5, and 1 hidden layers with 128 units each. We do not use any normalization layers, neither in MLPs nor in CNNs. During training, we sample completely random slices when training on isotropic materials (e.g. stone textures) by rotating a plane along all three axes. For more anisotropic structures like wood, we restrict random rotations along one axis, effectively minimizing the loss only along two axes and learning the third dimension implicitly. To efficiently compute texture values at many positions in a single forward pass, we implement $S$ using $1 \times 1$ convolutions. Training our model on a single exemplar takes a few hours on a single NVIDIA 2080ti GPU and learning an entire texture space takes up to several days depending on the dataset.

**Single exemplar** For efficiency, we perform many ablation studies in the single-exemplar setting. In order to draw robust conclusions we experimented with two very different exemplars: *wood* and *granite* (Figure 4 left). For comparisons we consider the following four variations of our framework: *GAN+VGG*: We replace our GramGAN style loss in $\mathcal{L}_G$ with VGG style loss (similarly to [21–23]). *NI*, Noise-at-Input: Instead of injecting noise as in Section 3.1, we inject it at the input layer of $S$. $\alpha$:1, $\beta$:0: We use a vanilla WGAN loss without any style loss. $\alpha$:0, $\beta$:1: We solely optimize for GramGAN loss without traditional WGAN term.

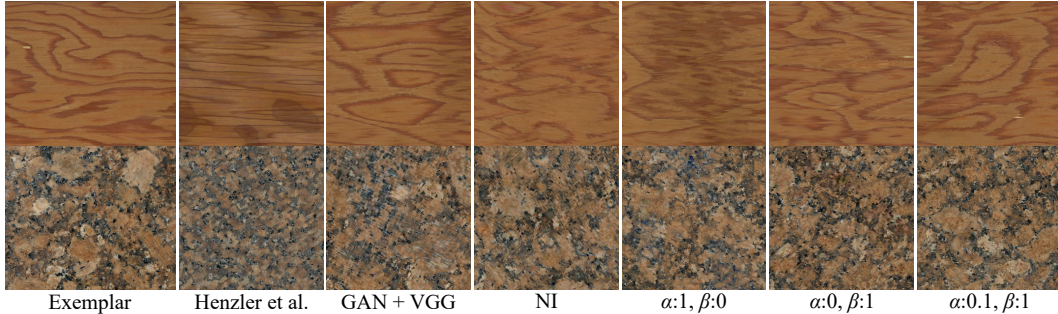

| Exemplar | Henzler et al. | GAN + VGG | NI | $\alpha$:1, $\beta$:0 | $\alpha$:0, $\beta$:1 | $\alpha$:0.1, $\beta$:1 |

Figure 4: Qualitative comparison on two exemplars. Our approaches (3rd to last column) produce significantly better results than the model proposed by Henzler et al. [5]. Moreover, our final model (last column) achieves more realistic results than various variants of it (3rd to 6th column).

Table 1: Quantitative comparison on the exemplars shown in Figure 4. We report SIFID $\mu_{\pm\sigma}$ ($\times 10^5$) (lower is better).

|  | Henzler et al. | GAN + VGG | NI | $\alpha$:1, $\beta$:0 | $\alpha$:0, $\beta$:1 | $\alpha$:0.1, $\beta$:1 |
|---|---|---|---|---|---|---|
| *wood* | $0.67_{\pm 0.42}$ | $0.19_{\pm 0.04}$ | $0.28_{\pm 0.04}$ | $0.29_{\pm 0.05}$ | $\mathbf{0.12}_{\pm 0.02}$ | $0.14_{\pm 0.02}$ |
| *granite* | $0.53_{\pm 0.03}$ | $0.31_{\pm 0.12}$ | $0.34_{\pm 0.29}$ | $0.28_{\pm 0.12}$ | $0.43_{\pm 0.2}$ | $\mathbf{0.21}_{\pm 0.10}$ |

In texture synthesis, the ground truth distribution is given by a single exemplar image and the quality of synthesized images is often evaluated by considering internal patch statistics. To quantitatively assess the differences between the approaches, we measure Single Image FID (SIFID) [15], a derivative of Fréchnet Inception Distance (FID) [30] tailored to quantify the distance between internal patch statistics of two images. For each method we create 50 random 3D textures (using different random seeds) and from each texture we synthesize a randomly oriented 2D slice at the resolution of the ground truth exemplar. We then compute SIFID for each of the 50 resulting sample images and report their mean and standard deviation in Table 1, with qualitative examples shown in Figure 4.

The examples show that all variants of our approach are capable of reasonably reproducing the target texture. However, we achieve significantly more plausible results with our proposed method (last column in Figure 4). In addition, we also train the model by Henzler et al. [5] on both exemplars and find that it produces significantly lower-quality results (second column in Figure 4). The numbers in Table 1 confirm these findings. While the results suggest that $\mathcal{L}_{\text{style}}$ is more important for the *wood* exemplar, WGAN loss is more effective on *granite*. However, note that solely optimizing for $\mathcal{L}_{\text{style}}$, i.e. $\alpha$:0, $\beta$:1, can cause artifacts that resemble Deep Dream [31] effects (tiny replicas of wood patterns when zoomed in Figure 4). For *NI* we use $\alpha$:0.1, $\beta$:1 and for *GAN+VGG* we chose the hyperparameters to approximately result in the same weighting as ours with $\alpha$:0.1, $\beta$:1. Interestingly, we found that *NI* results in significantly less stable adversarial training and we had to lower the learning rate by a factor of 5 to $1 \times 10^4$ to achieve convergence. For completeness we further evaluate a variant of our model where the entire noise vector $\eta$ is injected in each hidden layer instead of our binning approach described in Section 3.1. Due to the increase in network parameters implied by this variant, training this model takes significantly longer and performance is not competitive when optimizing using the same number of training iterations (on *granite* SIFID is $0.46_{\pm 0.14}$, which is better than [5] but worse than all other variants of our approach). In conclusion, using our GramGAN loss ($\alpha$:0.1, $\beta$:1) yields consistently competitive results without any artifacts for the given substantially different exemplars.

We show additional qualitative examples of our model in Figure 1 and 5, including a comparison to [5] on actual 3D renderings using the synthesized textures as diffuse albedo. The comparison in Figure 5 shows that [5] produces implausible artifacts in 3D for anisotropic textures. This is a cause of sampling only axis-aligned slices during training, while we sample all possible slice orientations. The renderings are raytraced using *mitsuba renderer* [32] modified to perform inference on our model for texture look-ups. The *cube* model is rendered using perfectly diffuse albedo and light source, while the *sphere* model rendering features a more realistic scenario with a glossy coating material and

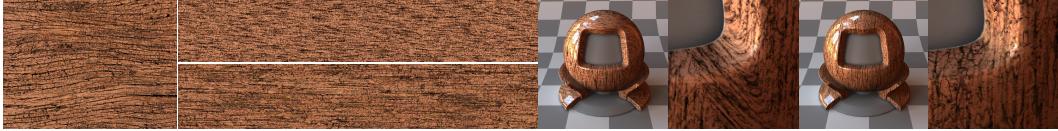

Figure 5: Qualitative comparison to Henzler et al. [5]. Left: exemplar. Middle: resynthesis Henzler et al. (top), ours (bottom). Right: 3D renderings Henzler et al. (left), ours (right).

Table 2: Quantitative evaluation of our framework in comparison to [5] and ground truth. We report $\mu_{\pm\sigma}$ for SIFID (lower is better), ALL (higher is better), and user study ranking (lower is better).

|                     | Ground truth            | Henzler et al.          | GramGAN (ours)          |
| ------------------- | ----------------------- | ----------------------- | ----------------------- |
| SIFID $\times 10^5$ | $\mathbf{2.46}_{\pm2.80}$ | $2.60_{\pm2.85}$        | $2.47_{\pm2.23}$        |
| ALL                 | $\mathbf{-294.85}_{\pm86.13}$ | $-319.02_{\pm70.61}$ | $-317.23_{\pm74.30}$ |
| rank from study     | $1.87_{\pm0.85}$        | $2.28_{\pm0.71}$        | $\mathbf{1.85}_{\pm0.80}$ |

an HDR environment map light source. The rendered images are high resolution, so please zoom in. Note that there is no rendering involved in training our models, since we learn diffuse textures only.

**Texture space** Next we report a quantitative evaluation of our conditional texture space model in comparison to the system proposed by Henzler et al. [5]. For this purpose we trained both models on a texture dataset consisting of 100 stone texture exemplars collected online. In this conditional setting, the input to both frameworks is an exemplar patch of resolution $128^2$ and the systems learn to resynthesize the input texture. For evaluation we synthesized texture samples of size $128^2$ from 22 reference input patches not seen during training, and measured their similarity to the references.

In Table 2 (first row) we report SIFID values averaged over the 22 samples generated by both frameworks. In addition, we also measured ground truth performance by sampling random crops from the respective training exemplars and compare them to the references. The numbers confirm that the outputs of our model are closer to the references in terms of SIFID. In addition, the comparison shows that SIFID for ground truth is not significantly lower. This can be explained by the fact that two random patches of the same texture exemplar image can look fairly different, especially for textures that feature very low-frequency variations. Note, however, that the computed SIFID numbers have rather high standard deviations due to the limited resolution of the texture images. In addition to SIFID we also measured average log-likelihood (ALL) reported in Table 2 (second row). An important property of generative models is to avoid producing artefacts and ALL has been proposed as a measure thereof [33], quantifying how probable generated samples are, given the ground truth density. ALL is typically estimated by using a Parzen window density estimate using a Gaussian kernel. Due to the stochastic nature of textures, we perform this evaluation over rather small kernel sizes to emphasize on the texture aspect. In particular, we start with a large kernel size and reduce its size until there is a significant difference between the ALL metric of the methods. To set the second parameter, Gaussian kernel bandwidth, we performed a grid search using the ground truth test set. Our evaluation shows that our method is superior also in terms of ALL, although not as significantly.

Next, we conducted a formal user study to evaluate both the perceptual quality of the generated samples and their similarity to the reference. For this purpose participants were presented with a reference texture patch and three candidate patches: the ground truth patch, a patch generated by our model, and a patch produced by [5]. The participants were asked to rank these according to their *similarity* to the reference patch. For sake of clarity, we elaborated similarity as *how likely that the candidate is an image of the same material as in the reference image, e.g. perhaps a different region/section of the same sample*. The presentation order of textures and candidates were randomized for each participant. We received rankings from 21 participants for 22 texture references. We report the average ranks of 462 rankings per method in Table 2 (third row). A paired Wilcoxon signed rank test confirms that the users prefer our results over [5] (with $p < 2 \times 10^{-11}$ at 5% significance level). Moreover, the difference between our results and ground truth is not statistically significant ($p = 0.56$), whereas ground truth is significantly better ranked than [5] ($p < 1 \times 10^{-10}$).

In Figure 6 we show a qualitative comparison of our texture space model on an exemplar texture patch with a texture significantly different than all training exemplars. This comparison shows that

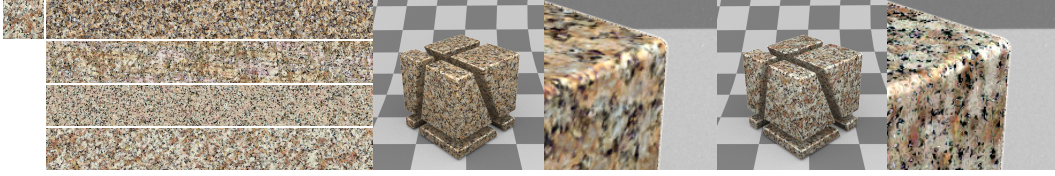

Figure 6: Qualitative results of conditional texture space model. Left: exemplar patch. Middle from top to bottom: Henzler et al., ours, ours extrapolated using VGG, ours extrapolated using $D$. Right: 3D renderings Henzler et al. (left) and ours (right), the latter looking more similar to the reference.

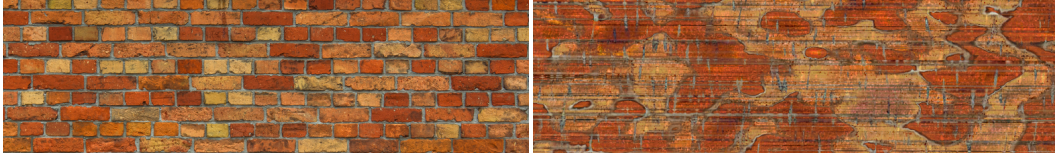

Figure 7: A failure case of our model, where the exemplar contains repetitive patterns. Left: exemplar, right: model output.

both [5] (first row) and our framework (second row) produce results fairly different from the input exemplar due to lacking extrapolation capabilities. Nevertheless, quantitative evaluation (SIFID over 50 random samples) indicate that our result ($2.04_{\pm 0.79}$) is significantly closer to the reference than [5] ($11.31_{\pm 0.79}$). Finally, we demonstrate that our extrapolation strategy proposed in Section 3.3 (forth row) further improves the output significantly ($1.46_{\pm 0.29}$). We also show an alternative of our extrapolation strategy (third row) that minimizes $\mathcal{L}_{\text{style}}$ using VGG instead of $D$. This experiment demonstrates that our discriminator features are much better suited than VGG features for style-based texture synthesis (SIFID: $1.79_{\pm 0.3}$). Note that whether a sample at hand is sufficiently close to the training data to omit the proposed extrapolation step is hard to estimate. One strategy would be to consider the distance of a sample at hand to the closest training exemplar in a suitable feature space. However, our experiments in this direction using Gram matrix distances in $D$ feature space do not provide clear insights. For instance, Gram matrix distance to the nearest neighbor is often higher for a held out test patch from a training exemplar than for a completely unseen texture.

**Limitations** Since our model synthesizes each texture value independently of its neighbors and the network has no direct access to the spatial input coordinates (the spatial information is hidden behind the noise frequencies), our technique cannot be used to generate structured textures such as repetitive patterns (see Figure 7) or complex stochastic structures (e.g. satellite images). Simply providing spatial information as additional network input unfortunately causes the model to loose diversity and memorize the training exemplars, as was also shown in [5].

## 5 Conclusions

In this work we presented GramGAN, an novel framework for 3D texture synthesis from 2D exemplars. Key to our method is a novel loss function that combines ideas from GANs as well as style transfer, to optimally guide a synthesis model. We demonstrate that our model significantly outperforms state of the art, with superior results compared to existing alternatives. An interesting future direction would be to extend our framework to non-diffuse albedos including specular and normal maps by leveraging differentiable renderers. Moreover, investigating extensions towards more structured and repeating textures yields great potential for future work. Finally, quantification of the generalization capability of our current model needs to be investigated further.

## Broader Impact

We present a novel framework for 3D texture synthesis that has broad applicability in computer graphics. It can be used to generate 3D and 2D textures, for instance in image manipulation, movie production, or game design. While it may be feasible to use dedicated models for each texture in

game design or movie production where a limited set of textures is used for each movie or video game, our conditional framework has the potential to greatly reduce artist work load. Our model has also great potential in medicine, for instance when training surgeons using virtual reality simulators. Content creation for such simulators, especially modeling different tissue materials, could benefit from our system. Besides generating realistic textures, our model would allow for the synthesis of animated textures or animated imagery by introducing an additional time dimension.

## Acknowledgments and Disclosure of Funding

We would like to thank the user study participants. Funding was provided by the Swiss Innovation Agency (Innosuisse) under MoCaFrame project (LS25540.1).

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
