[Supplementary Material]

# GramGAN: Deep 3D Texture Synthesis
# From 2D Exemplars

—

# Supplemental Material

**Tiziano Portenier[1], Siavash Bigdeli[2], Orcun Goksel[1]**
1: Computer-assisted Applications in Medicine, ETH Zurich, Switzerland
2: Swiss Center for Electronics and Microtechnology, Switzerland
{tiziano.portenier, orcun.goksel}@vision.ee.ethz.ch
siavash.bigdeli@csem.ch

In this supplemental material we show additional results and analysis of our framework for 3D texture synthesis. We first demonstrate the capability of our system to learn a continuous latent texture space when trained on a dataset consisting of diverse textures (Section 1). Next, we present more qualitative results that demonstrate the benefits of our approach compared to the system by Henzler et al. [1] and SinGAN [2], a 2D synthesis technique (Sections 2-4). In Section 5, we tabulate the network architectures of the convolutional neural networks used in our experiments. Finally, we present detailed results from our user study and all the images used therein in Section 6.

# 1 Latent space interpolation

Figure 1: Linear interpolation in latent space $z$. Top: first exemplar. Bottom: second exemplar. Middle: interpolated result. First and last square in each strip correspond to resynthesized exemplars.

## 2 Qualitative comparisons

Figure 2: Single exemplar setting. Left: exemplar (top), ours (middle), Henzler et al. (bottom). Right: ours (top), Henzler et al. (bottom). Note how our result is closer to the exemplar texture.

# 3    Comparison to 2D

Figure 3: Single exemplar setting. Here we show a qualitative comparison to the 2D synthesis framework SinGAN [2]. Left: exemplar (top), ours (middle), SinGAN (bottom). Right: ours (top), SinGAN (bottom). For texturing the 3D object with SinGAN, we first synthesize a 2D texture and sample this texture during rendering using the texture coordinates provided with the 3D model. While our 3D approach plausibly textures arbitrary surfaces, the 2D approach causes blur and distortions on the spherical surface. Note that the lighting setting is not exactly the same in both scenes, due to the use of a different renderer for bitmap texture lookups.

# 4  More results

Figure 4: Texture space setting. Here we demonstrate artifacts that occur due to non-axis-aligned slicing. If only axis-aligned slices are sampled during training ([1]), distortion artifacts occur along non-axis-aligned slices. Left: Henzler et al. axis-aligned cube (first and third row) and 45° rotated cube (second and fourth row). Right: ours axis-aligned cube (first and third row) and 45° rotated cube (second and fourth row). Note how ours produces more consistent results for both axis-aligned and non-axis-aligned geometry. See Figure 7 (second row) and Figure 8 (third row) for exemplar images.

Figure 5: Single exemplar setting. Here we show a example that violates the assumption behind a 3D texture. Given a grass training exemplar (top), our model learns a plausible 3D texture where random slices (middle) look similar to the target texture, although this setting would not be an ideal use of our proposed 3D approach.

# 5  Network architectures

Table 1: Network architecture of encoder $E$.

| layer | activaiton | shape in | shape out | kernel |
|-------|-----------|----------|-----------|--------|
| conv | LReLU | $128 \times 128 \times 3$ | $128 \times 128 \times 32$ | $3 \times 3$ |
| pool | – | $128 \times 128 \times 32$ | $64 \times 64 \times 32$ | $2 \times 2$ |
| conv | LReLU | $64 \times 64 \times 32$ | $64 \times 64 \times 64$ | $3 \times 3$ |
| pool | – | $64 \times 64 \times 64$ | $32 \times 32 \times 64$ | $2 \times 2$ |
| conv | LReLU | $32 \times 32 \times 64$ | $32 \times 32 \times 128$ | $3 \times 3$ |
| pool | – | $32 \times 32 \times 128$ | $16 \times 16 \times 128$ | $2 \times 2$ |
| conv | LReLU | $16 \times 16 \times 128$ | $16 \times 16 \times 256$ | $3 \times 3$ |
| pool | – | $16 \times 16 \times 256$ | $8 \times 8 \times 256$ | $2 \times 2$ |
| conv | LReLU | $8 \times 8 \times 256$ | $8 \times 8 \times 256$ | $3 \times 3$ |
| pool | – | $8 \times 8 \times 256$ | $4 \times 4 \times 256$ | $2 \times 2$ |
| conv | LReLU | $4 \times 4 \times 256$ | $4 \times 4 \times 256$ | $3 \times 3$ |
| pool | – | $4 \times 4 \times 256$ | $2 \times 2 \times 256$ | $2 \times 2$ |
| conv | LReLU | $2 \times 2 \times 256$ | $2 \times 2 \times 256$ | $2 \times 2$ |
| pool | – | $2 \times 2 \times 256$ | $1 \times 1 \times 256$ | $2 \times 2$ |
| dense | LReLU | 256 | 256 | – |
| dense | – | 256 | 32 | – |

Table 2: Network architecture of discriminator $D$.

| layer | activaiton | shape in | shape out | kernel |
|-------|-----------|----------|-----------|--------|
| conv | LReLU | $128 \times 128 \times 3$ | $128 \times 128 \times 32$ | $3 \times 3$ |
| conv | LReLU | $128 \times 128 \times 32$ | $128 \times 128 \times 64$ | $3 \times 3$ |
| pool | – | $128 \times 128 \times 64$ | $64 \times 64 \times 64$ | $2 \times 2$ |
| conv | LReLU | $64 \times 64 \times 64$ | $64 \times 64 \times 64$ | $3 \times 3$ |
| conv | LReLU | $64 \times 64 \times 64$ | $64 \times 64 \times 128$ | $3 \times 3$ |
| pool | – | $64 \times 64 \times 128$ | $32 \times 32 \times 128$ | $2 \times 2$ |
| conv | LReLU | $32 \times 32 \times 128$ | $32 \times 32 \times 128$ | $3 \times 3$ |
| pool | – | $32 \times 32 \times 128$ | $16 \times 16 \times 128$ | $2 \times 2$ |
| conv | LReLU | $16 \times 16 \times 128$ | $16 \times 16 \times 256$ | $3 \times 3$ |
| pool | – | $16 \times 16 \times 256$ | $8 \times 8 \times 256$ | $2 \times 2$ |
| conv | LReLU | $8 \times 8 \times 256$ | $8 \times 8 \times 256$ | $3 \times 3$ |
| pool | – | $8 \times 8 \times 256$ | $4 \times 4 \times 256$ | $2 \times 2$ |
| conv | LReLU | $4 \times 4 \times 256$ | $4 \times 4 \times 256$ | $3 \times 3$ |
| pool | – | $4 \times 4 \times 256$ | $2 \times 2 \times 256$ | $2 \times 2$ |
| conv | LReLU | $2 \times 2 \times 256$ | $2 \times 2 \times 256$ | $2 \times 2$ |
| pool | – | $2 \times 2 \times 256$ | $1 \times 1 \times 256$ | $2 \times 2$ |
| dense | LReLU | 256 | 512 | – |
| dense | – | 512 | 1 | – |

# 6 User study

Figure 6: User study images, palette 1. From left to right: reference, Henzler et al., ours, ground truth. Numbers show average user rank from our study (lower is better).

Figure 7: User study images, palette 2. From left to right: reference, Henzler et al., ours, ground truth. Numbers show average user rank from our study (lower is better).

Figure 8: User study images, palette 3. From left to right: reference, Henzler et al., ours, ground truth. Numbers show average user rank from our study (lower is better).

Figure 9: User study images, palette 4. From left to right: reference, Henzler et al., ours, ground truth. Numbers show average user rank from our study (lower is better).