[Reviews · NeurIPS 2020]

Review 1

Summary and Contributions: A novel method for synthesizing 3D textures given a 2D exemplar image. The method is composed of two parts: A 3D transformation model that maps a spatial 3D coordinate to a set of locations in frequency domain as an input noise vector; and a texture sampler that maps the combined spatial 3D coordinate and the noise vector to its RGB texture value. The texture interpretation results look very good. And there is clear improvement for the orientation of the texture for non-axis-aligned slices. Training the model on a single exemplar takes a few hours on a single NVIDIA 2080ti GPU, and learning an entire texture space takes up to several days depending on the dataset, which sounds reasonable considering the 3D volume that need to be optimized during the training.

Strengths: Here are a list of technical novelties I feel worth championing in the paper: 1. Two different ways to inject noise vectors into hidden layers of the sampler network, mimicking a styleGAN like architecture. In particular, the frequencies are partitioned into equal bins and each bin is injected to one hidden layer. 2. Use a combination of Gram matrix loss and GAN loss for training. The Gram matrix loss is minimized by L-1 distance (as opposed to L-2 distance), and uses a non-VGG feature extract network trained with images in the problem domain. 3. In each training iteration the model synthesizes 2D images by randomly slicing the 3D space. According to the author this is a crucial difference to previous axis-aligned slicing method [5], and produces less distortion artifacts along non-axis-aligned slices. 4. The method also learns a texture latent space given a set of many exemplars

Weaknesses: Only a limited number of texture examples are represented in the paper, and they are all pretty stochastic/irregular. I wonder if the author had experimented with more regular textures and observed where the method could break. Quantitative comparison on the exemplars shown significantly better SIFID scores in comparison to [5]. However, as the authors mentioned, such scores may not be an ideal measurement of model performance as GT is not significantly better, and the scores have high standard deviations due to limited resolution of the texture images. On the other hand, user study showed somewhat a clear preference for the proposed method over previous work [5], and no statistically significant differences between the proposed method and the ground truth.

Correctness: Sounds good to me.

Clarity: The paper is well-written. I have no problem of reading it.

Relation to Prior Work: Yes. The most relevant paper [5] is well discussed and compared in the submission.

Reproducibility: Yes

Additional Feedback: The rebuttal addressed my concern on limitation of the method on regular textures. I keep my original score ('Marginally above the acceptance threshold)


Review 2

Summary and Contributions: A novel texture synthesis model that maps input 3d coordinates to pixel values. It is used in the context of 3d texture synthesis, where every 2d slice of a 3d cube should be a valid texture. The work also introduces a loss combining GAN and perceptual gram matrix loss for better results. Also an extrapolation module is introduced for quick out of sample adaptation to novel textures.

Strengths: - it is an elegant approach to train a 2D GAN on slices from a generated 3D texture cube as fake images. This seems to scale well and to allow training much faster than 3D convolution approaches.

Weaknesses: - the paper seems too applied for the concrete application, and does not give any novel machine learning contributions, all loss functions and network design choices seem only moderately different than cited related texture generation approaches. Please emphasize more the novelty, e.g. key insights related to the 3D texture case. - some unclear details, see detailed comments in below sections for "clarity" and "related to prior work"

Correctness: yes

Clarity: overall well written It is unclear however, how rendering and lighting affect the GAN training: are the GAN generated texture slices read directly from the whole rendering engine, with lighting and other effects on top? (lines 265-267, 317-319) Or are they simpler 2d slices just from the output of the sample networks? Would be really good to clarify this, since the general NeurIPS scope covers usual 2D images, and the 3D rendering engines and their coupling to GANs are not self-explanatory - the word "frequency" used in section 3.1 seems misleading. There are no periodic functions used in the proposed model (unlike [18] where sine waves are explicitly added as parametrization to the latent space)

Relation to Prior Work: while important references like [17,18] are mentioned, it needs to be discussed more in light of the current paper -- I did not see them in the text referenced at all? These works can learn diverse textures from whole image sets (compare with line 48 of the current paper) and also learn a latent space of textures to be mixed (section 3.2 in the current paper). While [18] and related works do this for 2d images strictly, it is still very relevant to compare with the current proposed model for 3D textures.

Reproducibility: Yes

Additional Feedback: - the extrapolation fine tuning to novel textures seems too ad-hoc heuristically, and does not contribute to the main paper impact. Maybe section 3.3. can be shortened or left. and the paper can focus on more discussions or give more examples. - the texture examples seem too simple visually. Would be more impressive to include more complicated imagery, e.g. see examples in [17,18] , in order really to see how powerful the deep network is for texture synthesis. -Also discuss if texture patches of size 256x256 pixels of higher can be also learned with this architecture


Review 3

Summary and Contributions: This paper proposes a model that synthesizes 3D texture from a 2D exemplar image. The proposed method extends the framework of Henzler et al. by (1) formulating the task as a generative learning problem based on GAN, (2) introducing style loss to the generator based on Gram matrix computed using discriminator features, and (3) optimizing the generator architecture based on StyleGAN-style conditioning. The proposed method demonstrated non-trivial improvement in 3D texture synthesis over previous study (Henzler et al.).

Strengths: + The paper is generally well-written and easy to follow. + The practical improvement over the baseline (Henzler et al) seems to be non-trivial. + The ablation study is helpful to understand the source of improvements.

Weaknesses: - It would be great if authors give more background on the texture synthesis model (e.g., more details on sampler S, noise fields, etc.). Current draft is not self-contained and forces the readers (especially the ones not familiar with Perlin) to read [5] to get the full picture. - Although the whole framework has non-trivial improvements over [5], individual components are largely borrowed from existing literature. The entire synthesis pipeline is largely borrowed from [5], and the generator architecture basically follows the same principles in StyleGAN (i.e. the way to conditioning the generator). Some modifications such as random 2D slicing for loss computation and discriminator-based style loss are rather non-trivial, but technically still not strong especially considering that the idea of replacing pre-trained network using discriminator has been around over years (e.g, [A1] used discriminator feature to compute feature matching loss used to be computed using pre-trained VGG network). [A1] Wang et al., High-Resolution Image Synthesis and Semantic Manipulation with Conditional GANs, In CVPR, 2018

Correctness: Yes

Clarity: Yes

Relation to Prior Work: Yes

Reproducibility: Yes

Additional Feedback: Although there is no ground-breaking technical novelty, the proposed method reasonably extends [5] with GAN framework, and achieved non-trivial empirical improvement. ===== POST rebuttal ====== The rebuttal addressed my concerns. I keep my original rating.


Review 4

Summary and Contributions: The authors propose a framework for synthesis of 2D and 3D textures from a 2D texture exemplar. They employ an adversarial objective along with perlin noise to synthesis corresponding textures. The authors demonstrate the their approach on several textures to motivate the method. Additionally, adequate comparisons are provided to demonstrate state of the art performance. Further, the ablations provided motivate the need for each of the components introduced.

Strengths: The paper is well written with adequate attention to detail. 1. The authors present appropriate motivation for the problem and its use cases. Particularly, the problem of generating 3D textures using 2D exemplar has wide variety of applications in graphics. 2. The formulation is adequately motivated and is intuitive and easy to understand. The transformation of the Perlin noise and sampling strategy employed is described clearly. 3. The proposal of combining the GAN loss and style loss is an elegant idea. Particularly, as the authors point out, the discriminator is specifically trained for the task of identifying the textures, which could possibly cause the network to learn more relevant texture features than features from a pretrained networks. 4. The results demonstrated are of high quality and are presented with lighting conditions as well unlike prior art, which highlights result quality. 5. Ablations are provided to motivate the need for each of the loss terms. Particularly the advantage of using the discriminator to extract style features over pretrained VGG network is demonstrated effectively in the experiments. 6. The authors present comparison to prior state of the art and demonstrate the effectiveness of the proposed approach. 7. The experimental analysis section is covered in great detail and adequate insights are provided regarding the observed results in terms of both the qualitative and quantitative performance. 8. The supplementary section provides a large number of examples demonstrating the effectiveness of the approach. The result quality on texture interpolation is also very promising. 9. The user study design is explained in great detail. Particularly, the choice of scoring employed and the scale of the study seems appropriate to draw statistically significant results. 10. The proposed solution is also potentially useful outside the texture synthesis application to any works that use style transfer, since it presents an interesting way of evaluating style features.

Weaknesses: Despite being well written with adequate insights, the manuscript would make for a stronger submission by addressing some of the concerns below, 1. The work demonstrates high quality results however, the presented framework is an incremental extension of [5]. Particularly, the approach augments the loss in [5] with an adversarial component and feeds the conditioning at each layer similar to StyleGan. 2. The experimental evaluation presented in significantly rigorous as compared to the most prominent prior art. Particularly, the authors provide results only on 2 texture classes. 3. Lines 174: The authors make a design choice to split the frequencies into different sized bins and feed it to different layers. The authors claim that this is beneficial since this reduces the number of parameters. Is this beneficial in terms of performance? or training speed? . Where exactly are the benefits of this choice manifested? An experiment to support this design choice would be instructive. Particularly, demonstrating an ablation with feeding the full vector at each layer with A being (m x n) instead of (m x n/4). 4. The authors provide quantitative evaluation highlighting that the generated patches have adequate similarity to input exemplar. However, as in the claims of lines 51-55, the authors present no evaluation regarding the diversity of the synthesized patches. Particularly, how do the authors make sure that the quality is not improved at the cost of diversity? 5. Comparison is provided only to [5] and no other previous methods. Although [5] demonstrates state of the art performance over several previous method already, it is important to highlight the performance of the presented framework in the context of other similar texture synthesis methods like [a,b,c]. 6. Lines 193: A more detailed treatment of "sufficiently similar" texture would provide better insights regarding the generalizability of the framework. Particularly, how does one measure the similarity of the textures. A simple study would be, to demonstrate the generalization performance as a function of Gram Matrix distance of a test texture to the nearest neighbor training texture. This will also lend credence to the fact that the proposed framework beats prior art in terms of quality. [a] Dmitry Ulyanov, Andrea Vedaldi, and Victor Lempitsky. Improved texture networks: Maximizing quality and diversity in feed-forward stylization and texture synthesis. In CVPR, 2017 [b] Dmitry Ulyanov, Vadim Lebedev, Andrea Vedaldi, and Victor S Lempitsky. Texture networks: Feed-forward synthesis of textures and stylized images. In ICML,2016 [c] Ning Yu, Connelly Barnes, Eli Shechtman, Sohrab Amirghodsi, and Michal Lukac. Texture Mixer: A network for controllable synthesis and interpolation of texture. In CVPR, 2019

Correctness: The authors build their claims on the work of [5]. The assumptions presented are valid and the experimental evaluation is appropriate to evaluate the proposed method. Although the experimental section can be made more rigorous by incorporating additional comparisons and ablations as mentioned above.

Clarity: The paper is well written with great attention to detail and adequate insights regarding each design decision and qualitative observations. The motivation and related work section build necessary motivation and context for the work. The formulation is presented in detail and is easy to understand. The presented experiments logically follow from the design choices.

Relation to Prior Work: The work has been adequately placed in the context of prior art. Particularly most relevant works for texture synthesis and texture interpolation have been cited.

Reproducibility: Yes

Additional Feedback: %%%% Post Rebuttal: The authors have addressed most of the concerns raised by the reviewers in the rebuttal. To that end, I am inclined to retain my original score for acceptance

[Author Response · NeurIPS 2020]

We thank the reviewers for their valuable feedback, and we address raised questions and comments below.

**R1: Discussion of limitations, repetitive patterns.** Thank you for pointing to the interesting question regarding regular textures. Indeed Perlin noise based techniques by design cannot synthesize repeating textures, since the texture is the result of sampling an underlying infinite noise field, designed to capture randomness in natural objects. It is therefore well suited for natural 3D textures. We will discuss these limitations explicitly in our revised text.

**R2: Machine learning contributions.** A main novelty is our proposed loss function, with *discriminator features for style loss* leading to better results than existing approaches (GAN loss, VGG style loss) for texture synthesis. Our loss enables training *a conditional GAN using an unconditional discriminator*. This is crucial in texture synthesis and related fields: This not only relaxes constraints on the discriminator (therefore providing better gradients to the generator), but it also effectively avoids practical difficulties with a conditional discriminator. This can be imagined, e.g., considering an exemplar image where the texture in the top left differs from the texture at the bottom right. We evaluated the conditional alternative extensively and we will emphasize this aspect in more detail in the text. We expect our novel loss to be applicable beyond texture synthesis; potentially in related fields such as style transfer.

**R2: Rendering during training.** In this work we learn only diffuse textures. An explicit rendering step is thus omitted and the loss is computed directly on the synthesized slice. After training, as an application demo we use the learned textures in a rendering framework (lines 265-267) under various lighting conditions. We will clarify this in the text.

**R2: Use of word "frequency".** By frequency we refer to how we sample the noise fields (which we do in a periodic manner), not the noise signal itself. We will better define this in the text.

**R2: Discussion of references [17,18].** We mentioned these references in Section 2 (line 100-101). Similarity of these approaches to ours is a noise field being transformed to a texture, enabling infinite and seamless synthesis. However, in [17,18] the transformation is parametrized with CNNs unlike ours. While this is efficient in 2D, it is not feasible in 3D. The most important difference is their restriction to 2D as well as our novel loss function. We will discuss these references comparatively and add a comparison to the supplemental material.

**R2: Relevance of Section 3.3 on texture extrapolation.** We believe that our extrapolation strategy is important in practice since it reduces training time substantially for adding any new samples to a dataset, which is common in production. Although we would prefer to keep this section in the main text, if there is consensus that it does not strengthen the submission, we will gladly move it to supplementary material.

**R2: Complexity of shown examples.** Many common texture synthesis examples are not 3D volumetric materials. Our application domain of 3D textures somewhat limits our choice of examples, e.g., to volumetric materials like stone and wood. We will include additional and more diverse examples in the supplemental material.

**R2: Training patch size.** The training patch size (which defines the receptive field) can be chosen arbitrarily, i.e., our system can be trained on larger patches ($256^2$ or higher). Such choice determines the range of frequencies in the training data captured by our network. We will mention this in the text.

**R3: Not self-contained.** Thank you for pointing this out. We will gladly include the missing details in the text.

**R3: Novelty in replacing VGG with discriminator.** It is true that perceptual loss on discriminator features has been proposed earlier, we will discuss this explicitly in the revised text. This concept has been applied in image reconstruction and image translation tasks, where an input is mapped to its ground truth. However, we do not aim to reconstruct the content of our input, but to reproduce its style. Thus we propose a novel style loss based on discriminator features.

**R4: Design choice to split frequencies.** With this we are able to reduce the training time by roughly 20% without noticeable degradation of output image quality. We will gladly include an ablation experiment.

**R4: Evaluation of diversity.** We would like to emphasize that our model provides diverse outputs, by construction, similarly to [5] (due to sampling and combination of infinite noise fields). It is true that we do not quantitatively evaluate diversity in an isolated manner; however, we demonstrate diversity qualitatively by synthesizing large stripes, several times larger than the training exemplars. In addition, we report FID, which was shown in the literature to correlate well with both quality and diversity (although not a perfect metric). We believe that our quantitative evaluation with synthesis of 50 slices that are significantly larger than the receptive field/training patch size demonstrates diversity.

**R4: Comparisons to 2D texture CNNs ([a,b,c]).** We already discussed [a,b] in the original submission. Thank you for pointing out reference [c], which we will gladly include. A visual comparison to demonstrate the limitations of 2D approaches in comparison to ours is an excellent idea, thank you for suggesting this. In the supplementary material, we will gladly provide such a comparison, which will strengthen our manuscript.

**R4: Generalization w.r.t. Gram matrix distance.** Thank you for your suggestion. This is indeed an interesting experiment. We will investigate such relationship and report our findings in the revised version.

[Meta-Review · NeurIPS 2020]

Reviewers were unanimous in their appraisal of the submission's clarity and that the technical achievement was non-trivial and useful to the community, although views differed on the nature of what that achievement actually was (R2 contends that the "machine learning" contributions are thin, while this is disputed in the rebuttal; R2 did not update their review in response as of this writing). Regardless of the above, the paper's evaluations are strong enough for me to unreservedly recommend acceptance. I would nonetheless urge the authors to incorporate feedback of R3 and R4 in the camera-ready.